# Automated Super-Network Generation for Scalable Neural Architecture Search

**J. Pablo Muñoz**[1,2] **Nikolay Lyalyushkin**[2] **Chaunte Lacewell**[1,2] **Anastasia Senina**[2]
**Daniel Cummings**[1, 2] **Anthony Sarah**[1, 2] **Alexander Kozlov**[2] **Nilesh Jain**[1, 2]

[1]Intel Labs
[2]Intel Corporation

**Abstract**   Weight-sharing Neural Architecture Search (NAS) solutions often discover neural network architectures that outperform their human-crafted counterparts. Weight-sharing allows the creation and training of super-networks that contain many smaller and more efficient child models, a.k.a., sub-networks. For an average deep learning practitioner, generating and training one of these super-networks for an arbitrary neural network architecture design space can be a daunting experience. In this paper, we present BootstrapNAS, a software framework that addresses this challenge by automating the generation and training of super-networks. Developers can use this solution to convert a pre-trained model into a super-network. BootstrapNAS then trains the super-network using a weight-sharing NAS technique available in the framework or provided by the user. Finally, a search component discovers high-performing sub-networks that are returned to the end-user. We demonstrate BootstrapNAS by automatically generating super-networks from popular pre-trained models (MobileNetV2, MobileNetV3, EfficientNet, ResNet50 and HyperSeg), available from Torchvision and other repositories. BootstrapNAS can achieve up to 9.87× improvement in throughput in comparison to the pre-trained Torchvision ResNet-50 (FP32) on Intel Xeon platform. Our code is available at `https://github.com/jpablomch/bootstrapnas`

## 1 Introduction

Neural Architecture Search (NAS) solutions attempt to identify the architectures with the best performance from a search space of possible neural network configurations. In the past few years, weight-sharing NAS approaches have produced outstanding results (Cai, Gan, et al., 2020; Yu and Huang, 2019b). These approaches build a super-network from which smaller and, in some cases, more efficient child models, a.k.a., sub-networks, can be extracted. Unfortunately, generating a super-network, either from scratch or from an existing pre-trained model, can be a challenging experience. One has to create the main data structure for the super-network, and include mechanisms for activating, extracting, forward and backward passing on selected sub-networks. This process is repeated when a new super-network needs to be generated for a different search space.

In this paper, we present BootstrapNAS, a software framework for scalable super-network generation and training. The BootstrapNAS approach focuses on automatically deriving a super-network from an existing network architecture. This paper extends our short workshop paper (Muñoz et al., 2021). This work provides the following contributions to the AutoML community:

I. A software framework that automates the generation and training of super-networks from pre-trained models, and the subsequent search for high-performing sub-networks.

II. Highly extensible APIs that allow developers and researchers to incorporate their own methods for training super-networks and discovering high-performing models.

III. A hardware-aware solution that incorporates measurements collected at a target hardware during the sub-network search stage, or by using trained predictors for these measurements. Users can create their own performance estimators and provide them to the BootstrapNAS' API.

## 2 Related Work

There has been significant progress in Neural Architecture Search (NAS) in the past few years. At the core of all NAS solutions is the exploration of a *search space*, guided by a *search strategy* and a *performance estimation strategy* (Elsken et al., 2019). In this paper, the focus is on NAS approaches that avoid training several individual models in separate training events since this is impractical for large neural architecture design spaces. The emphasis is on weight-sharing NAS approaches, e.g., (Bender et al., 2018; Cai, Gan, et al., 2020; Guo et al., 2020; G. Li et al., 2019; Liu et al., 2018; Pham et al., 2018; Yu, Jin, et al., 2020), and in particular on those that are or can be made hardware-aware during the sub-network search stage, e.g., (Cai, Gan, et al., 2020; Yu and Huang, 2019a).

Several frameworks have been proposed to automate the generation of the NAS search space and identify high-performing models. ModularNAS (Lin et al., 2021) provides a unified interface that allows the user to implement several state-of-the-art NAS methods, including super-network training and super-network-based search. The *Retiarii* framework (Zhang et al., 2020), a component of Microsoft's Neural Network Intelligence (NNI) (Microsoft, 2021) allows the user to design the model space, and then apply a search strategy. NASLib is a library that requires minimal coding efforts to define or reuse an existing search space (Ruchte et al., 2020), enabling researchers to quickly test their NAS methods on well-known search spaces. Another relevant work is Fast Neural Network Adaptation (FNA) (Fang et al., 2020), in which the architecture and parameters of a high-performing pre-trained backbone are used to produce alternative architectures, i.e., sub-networks, for different tasks, e.g., detection and segmentation. Other frameworks have been proposed in the past to define a NAS search space. For instance, DeepArchitect (Negrinho and Gordon, 2017; Negrinho, Patil, et al., 2019) provides a custom language for representing the search space, and separately trains a set of models from scratch. In contrast to these frameworks, BootstrapNAS automatically generates the search space from a given pre-trained model, automating the construction of a super-network and minimizing the required coding from the user.

**Limitations**. Generalizing the automated generation of NAS super-networks from arbitrary pre-trained models is a great challenge that we confront as an iterative process. Future versions of BootstrapNAS will improve its capabilities allowing for more users to benefit from this software framework. Currently, there might be models that are not suitable for this approach, either because they have been efficiently optimized and it is difficult to discover high-performing sub-networks, or because they contain custom operators that are still not supported in BootstrapNAS, hence preventing the generation of their super-networks. Another limitation might be the complex configurations and setting of hyperparameters that might be required for some models. In some cases, BootstrapNAS' automatic detection of potential elastic layers might result in a vast search space affecting training and searching times. To address this concern, BootstrapNAS allows for a manual setting of the super-network's elasticity hyper-parameters. (Elasticity is defined in section 3.1).

**Societal Impact**. Applications that use deep learning models are ubiquitous. Unfortunately, they come with a significant environmental cost. For instance, the training and deployment of these models are associated with a significant increase in $CO_2$ emissions. We believe that research in efficient solutions for model compression, e.g., BootstrapNAS, will result in energy savings by producing smaller models with a reduced carbon footprint compared to their less efficient hand-crafted counterparts.

## 3  Framework Overview

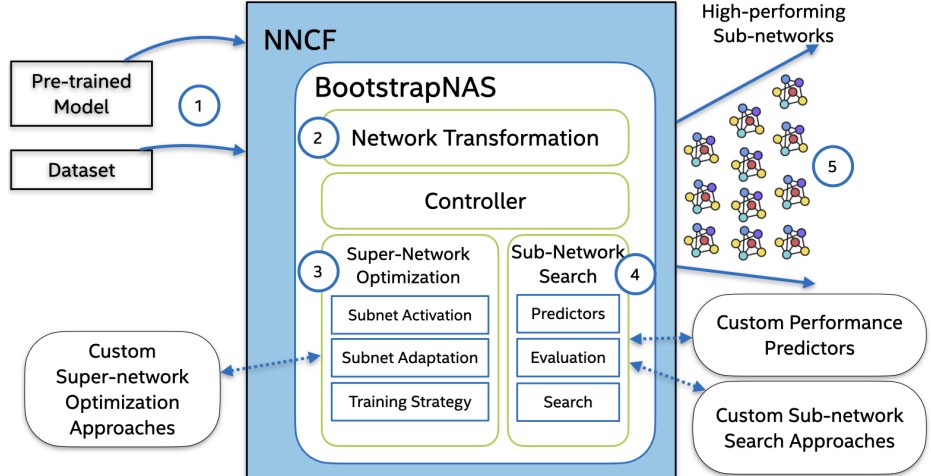

Figure 1: BootstrapNAS architecture within the NNCF framework. (1) BootstrapNAS takes as input a pre-trained model and a dataset. (2) It transforms the model into a super-network. (3) It then trains the super-network by activating sub-networks, either randomly or based on a particular training strategy. (4) Once the super-network has been trained, BootstrapNAS searches for efficient sub-networks, and (5) returns to the user a set of sub-networks that satisfy the specified objectives. BootstrapNAS is highly extensible allowing for the incorporation of alternative approaches for training super-networks and for searching for high-performing sub-networks.

BootstrapNAS is a software framework for the automated generation of weight-sharing super-networks for Neural Architecture Search (NAS). It is being developed within the Neural Network Compression Framework (NNCF) (Kozlov et al., 2020). As illustrated in Figure 1, (1) BootstrapNAS takes as input a pre-trained model, $m$, and a dataset, $D$, from the user. (2) It then, transforms $m$ into a super-network, $\Omega$. (3) The super-network is trained using one of the available training strategies. (4) Once the super-network has been trained, BootstrapNAS searches for high-performing sub-networks which are returned to the user at (5). In the following sections, we describe each stage in more detail.

Table 1: Notation

| | | | |
|---|---|---|---|
| $\Omega$ | Super-network | $L^\Omega$ | Set of layers of $\Omega$ |
| $a_i$ | Sub-network $i$ | $l_i^\Omega$ | Layer $i$ of $\Omega$ |
| $a_{min}$ | Minimal sub-network | $L^i$ | Set of layers of $a_i$ |
| $a_{max}$ | Maximal sub-network | $l_j^i$ | Layer $j$ of $a_i$ |
| $m$ | Pre-trained model | $L_s^i$ | Set of *static* layers |
| $A$ | Set of all sub-networks | | of $a_i$ |
| $A_o$ | Set of Pareto-optimal | $L_e^i$ | Set of *elastic* layers |
| | sub-networks | | of $a_i$ |

### 3.1  Automated Super-Network Generation

BootstrapNAS automatically generates a super-network, $\Omega$, from a pre-trained model, $m$. A weight-sharing super-network is similar in its structure to other neural networks. The difference is that in a super-network, it is possible to activate different configurations for some of its layers, allowing for the extraction of child models, a.k.a. sub-networks. The literature in this topic often uses the term **elasticity** to refer to the capability of certain operations in a neural network to be configured with different values (Cai, Gan, et al., 2020). The objective at this stage in BootstrapNAS is to make a few selected layers *elastic* in $\Omega$, hence allowing the possibility of manipulating smaller

weight-sharing sub-networks. Currently, BootstrapNAS automatically generates super-networks applying elasticity to three dimensions of the network: depth, width and kernel size.

Converting $m$ into a super-network $\Omega$ is accomplished by using the capabilities of the Neural Network Compression Framework (NNCF). NNCF traces the pre-trained model's graph and wraps each candidate operator with pre- and post-operations effectively making an operator *elastic*. This step is illustrated in Figure 2. During this stage, the super-network, $\Omega$, starts as a copy of $m$ and then selected layers are transformed into *elastic* layers. The set of layers in $\Omega$, $L^{\Omega}$, is composed of two subsets of layers. One subset of *elastic* layers, $L_e^{\Omega}$, and another subset of *static* layers, $L_s^{\Omega}$. Static layers are shared by all the sub-networks, while elastic layers, $L_e^{\Omega}$, might not always be shared by all sub-networks.

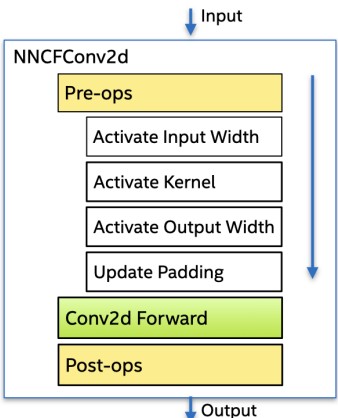

Figure 2: BootstrapNAS uses NNCF capabilities to make an operator *elastic* by wrapping it with pre- and post-operations. As illustrated in this figure, a *Conv2d* operator is wrapped with pre-ops that affect its input and output, effectively making it *elastic*.

Once an operator has been wrapped, BootstrapNAS can automatically activate different configurations for each layer. Each elastic layer has various possible values for its properties. For instance, the layer could allow several values for its number of channels. e.g., {512, 256, 128}, or kernel sizes in the case of convolutional layers. Since the super-network, $\Omega$, is created from the pre-trained model, $m$, the maximum value for a property in a layer $l_i^{\Omega}$ of $\Omega$ will be equal to the original value for the same property in the same layer, $l_i^m$ of $m$.

Among the many sub-networks, $a_i$, contained in the super-network $\Omega$, there are two sub-networks that are of particular importance: the *maximal* sub-network $a_{max}$, which has a configuration of layers, and the values for its properties equal to those in the given pre-trained model $m$, and hence the transformation from $m$ to $\Omega$ must guarantee that the maximal sub-network $a_{max}$ results in the same accuracy (within a small margin of error) as the original model $m$ on a dataset $D_{val}$. That is, $Cost(a_{max}, D_{val}) \cong Cost(m, D_{val})$. Otherwise, this is an indication that an error occurred during the super-network generation step. The other sub-network that is of particular importance is the *minimal* sub-network, $a_{min}$. The configuration of this sub-network has the minimum possible value for each property in every elastic layer. More details on how BootstrapNAS enables layer elasticity can be found in Appendix A.

## 3.2 Super-Network Training

Once the super-network has been automatically generated, BootstrapNAS trains it using one of the available training strategies. The literature on super-network training contains several training strategies, e.g., *progressive shrinking* (Cai, Gan, et al., 2020) or *single stage* training (Yu, Jin, et al., 2020). BootstrapNAS uses progressive shrinking by default. Using this strategy, the training of the

super-network occurs in multiple stages, and in decreasing order for each of the properties of elastic layers. For instance, it first activates different elastic depth configurations in decreasing order, then it activates elastic depth and elastic width in decreasing order, and so on. The training schedule is derived from the training schedule of the source model, which simplifies the requirements for the user. Knowledge distillation can be applied during training (Hinton et al., 2015). The soft labels from $m$ or from the maximal subnetwork, $a_{max}$, can be used to compute the loss of the sampled sub-networks. Using the soft labels of $a_{max}$ is termed *inplace distillation* (Yu and Huang, 2019b)

A simple *single stage* training can be obtained from within the progressive shrinking strategy by activating all the elastic dimensions at once and allowing for the activation of all their possible values at the layers' properties. Then, a subset of sub-networks are sampled at random, and their gradients are aggregated and used to update the weights of the super-network. A more complex single stage training, e.g., the *sandwich* rule (Yu and Huang, 2019b) requires, at each training step, the sampling of the maximal, $a_{max}$ and minimal $a_{min}$ sub-networks, together with other $n$ sub-networks sampled at random. As with other training strategies, the weights are aggregated and used to update the super-network. BootstrapNAS' API is highly extensible allowing advanced users to implement their own training strategies.

### 3.3 Sub-Network Search

Once the super-network optimization training stage has finished, the next step is to search and return $k$ sub-networks from the set of Pareto-optimal sub-networks, $A_o$, ($A_o \subseteq A$), to the user. We explain below how $A_o$ is obtained. As default, $k = 1$, that is, a single sub-network from the Pareto set that outperforms the original model, $m$, is returned to the user. To select $k$ networks from $A_o$, BootstrapNAS favors sub-networks that have an accuracy similar to $m$ (with some tolerance) but improve efficiency, e.g., minor drop in accuracy but significant improvement in latency.

The search of Pareto-optimal sub-networks, $A_o$ can be approached as a multi-objective optimization. The multi-objective goal is to,

$$\text{minimize} f_1(\mathbf{x}), \ldots, \ \text{minimize} \ f_n(\mathbf{x}), \mathbf{x} \in \mathcal{X}, \tag{1}$$

for a given set of $n$ objective functions $f_1 \colon \mathcal{X} \to \mathbb{R}, \ldots, f_n \colon \mathcal{X} \to \mathbb{R}$, where $\mathbf{x}$ is a member of an objective decision space $\mathcal{X}$ (Emmerich and Deutz, 2018). Multi-objective evolutionary algorithms (MOEAs), specifically Pareto-based MOEAs, are well suited for sub-network search problems since they can operate easily on the discrete variable types given by the super-network elastic parameter space. Many are designed to evolve towards sets of Pareto optimal solutions that have a diverse spread across the Pareto front (objective trade-off solution region). As a default search algorithm, BootstrapNAS uses NSGA-II (Deb, Pratap, et al., 2002). NSGA-II uses a generational loop process that evolves a population of individuals (e.g., sub-networks) using crossover and mutation operations and then ranks the different individuals in the population using a non-dominated sorting with crowding distance criterion to produce a diverse set of Pareto-optimal solutions. With NSGA-II, a user is provided with a wide and diverse spread of sub-network options across the objective space.

BootstrapNAS uses by default the multi-objective optimization capabilities of *Pymoo* (Blank and Deb, 2020), which enables the smooth application of NSGA-II and other search algorithms. For instance, a user can randomly sample sub-networks and then use the performance measurements of a particular sub-network as a reference point to other algorithms. In the case where only a specific region of the objective space is desirable, RNSGA-II (Deb and Sundar, 2006) offers a solution to direct the search. As illustrated in Figure 3, BootstrapNAS can use the output of the user's random search and a particular accuracy target to explore a region of the model space. Reference point based multi-objective optimization approaches allow one or more reference points to be defined in the objective space and where the ranking of solutions are calculated by the euclidean distance to each reference point. In our example, RNSGA-II requires fewer evaluations than NSGA-II. While NSGA-II

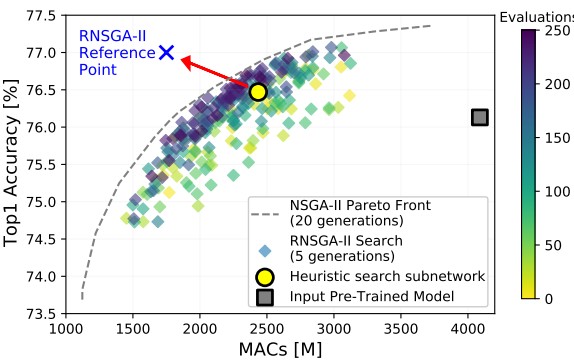

Figure 3: Reference point genetic algorithm search progression. The performance measurements of a sub-network, e.g., from the application of random search or hand-picked by the user, can be used as a starting point for a more advanced *directed* search using RNSGA-II (Deb and Sundar, 2006)

.

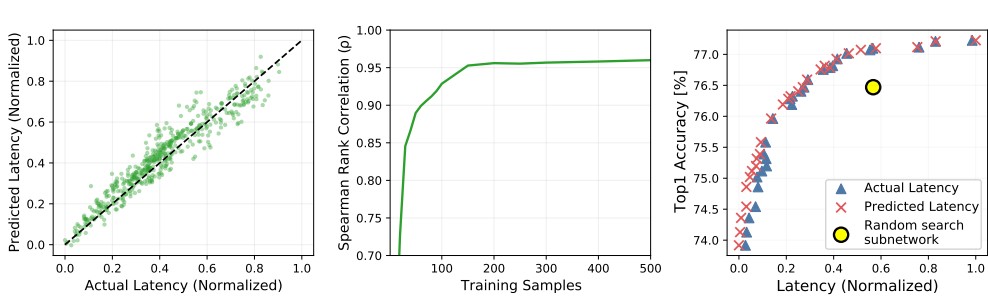

Figure 4: Latency predictor training examples. The left plot shows the predicted versus actual latency after a ridge regression model is trained (test=500 samples, train=500 samples). The center plot shows how the Spearman's rank correlation coefficient improves with training samples. The right plot presents a comparison of predicted versus measured latency from optimal sub-networks identified during a NSGA-II search using the simple example latency predictor. Although error is introduced when using predictors, the discovered sub-networks tend to outperform the ones found with random search.

can take a significant number of evaluations to produce good results, it is easily applied to any super-network, does not require reference points, and shows consistent evolutionary progressions towards finding better sub-network solutions.

**Performance Estimation** The search stage in super-network-based NAS solutions often requires a significant amount of time depending on the size of the search space and the selected search strategy, mostly because of the cost of measuring the performance metrics of the sub-networks. One common approach is to sample and evaluate a subset of sub-networks on metrics, such as accuracy and latency, and use these measurements to train predictors since search spaces can often reach a very large number of possible sub-networks configurations, e.g., $10^{19}$ for MobileNetV3 (Cai, Gan, et al., 2020). Lookup tables are another common form of predicting latency which aggregates delay by layer operation to approximate the full delay of the sub-network of interest (Cai, Zhu, et al., 2019). Once these predictors have been created, the search stage can take a relatively short amount of time. Other metrics that are commonly used to approximate model size and complexity are model parameter counts and multiply-accumulates (MACs). The evaluation of these metrics tends to take less time. The best practice when using predictor approaches is to perform a final validation measurement on the best candidate sub-networks since predictors introduce some level of inaccuracy depending on how they were trained. The rationale is that to search across tens of

thousands of sub-network options quickly, you can just use a well-behaved predictor, which can be trained in less than a thousand samples. BootstrapNAS allows any predictor modeling approach to be used jointly with the sub-network search component. While we use ridge regression for our example, another comprehensive work (Lu et al., 2021) covers the use of different predictor models (e.g., MLP, RBF, decision trees).

To illustrate the use of predictors, we randomly sample subnetworks from the BootstrapNAS Resnet-50 model and measure their latency. In Figure 4, we illustrate that the latency predictor can achieve a Spearman's rank correlation coefficient $\rho > 0.95$ in as few as 200 training samples in this example. Next, we perform 20 generation (population=50) NSGA-II search using the latency predictor and measure the actual latency on the best Pareto-optimal sub-networks at the end of the search. When viewed in the latency and accuracy objective space as in Figure 4, the resulting Pareto front sub-network solutions behave as expected. In the BootstrapNAS framework, predictors can be utilized on one or both objectives during multi-objective search, and the end-user has the flexibility to implement their own estimators and pass them as arguments during the search stage.

## 4 Experiments

The main goal of these experiments is to demonstrate the generalization capabilities of BootstrapNAS for automated super-network generation, training and search for high-performing sub-networks. We generated several super-networks from popular models using BootstrapNAS. Some of these super-networks might be suitable for the application of other model compression techniques, e.g., quantization, or longer super-network training to further improve their performance.

In the process of implementing the BootstrapNAS approach, we started by extending the code from (Cai, Gan, et al., 2020) to generate super-networks from Torchvision's[1] ResNet-50 (He et al., 2016) model trained with Imagenet (Deng et al., 2009) and HyperSeg Lite trained with the VFX dataset (Rhodes and Goel, 2020). In our second development iteration, we implemented BootstrapNAS' current scalable open-source API and generated super-networks for ResNet-50 trained with CIFAR-10 (Krizhevsky et al., 2009), MobilenetV2 (Sandler et al., 2018) (CIFAR-10 and Imagenet), MobilenetV3 (Howard et al., 2019) trained with Imagenet, and EfficientNet (Tan and Le, 2019) trained with CIFAR-100. Sub-network search uses NSGA-II as default with crossover rate of 0.9, mutation rate of 0.02, a population of size 50, 3000 evaluations for CIFAR-trained models, and 1000 evaluations for Imagenet-trained models. These values were chosen based on our optimization ablation studies.

**Super-Networks from Models Trained with CIFAR-10**. To demonstrate BootstrapNAS' automated generalizable super-network generation capabilities, we selected two models: ResNet-50 and MobileNet-V2, both trained with CIFAR-10 from (Phan, 2021). As illustrated in Figure 5 (two plots on the right), several sub-networks discovered by BootstrapNAS outperform the given pre-trained models. We highlight two sub-networks for each super-network (BootstrapNAS A-RC, B-RC, A-MC, and B-MC). Table 2 describes the reduction in required MACs by these sub-networks while either maintaining the accuracy of the baseline model, or with a minor drop in accuracy.

**Super-Networks from Models Trained with CIFAR-100**. We also used BootstrapNAS to generate a super-network from EfficientNet-B0 (CIFAR-100). This model has a top 1 accuracy of 87.02% after transfering from Imagenet weights (*EfficientNet-PyTorch* 2021). As described in Table 3, BootstrapNAS discovered a sub-network that reduces the number of MACs by 12% with a minimum drop of accuracy from 87.02% to 86.89%. This improvement is achieved with only 20 epochs of super-network training on the elastic depth dimension.

**Super-Networks from Models Trained with Imagenet**. Three super-networks were generated from Resnet-50, MobilenetV2, MobileNetV3, all pre-trained models available at Torchvision and trained with Imagenet. In all three cases, BootstrapNAS discovered sub-networks that are more

---

[1]https://pytorch.org/vision/stable/index.html

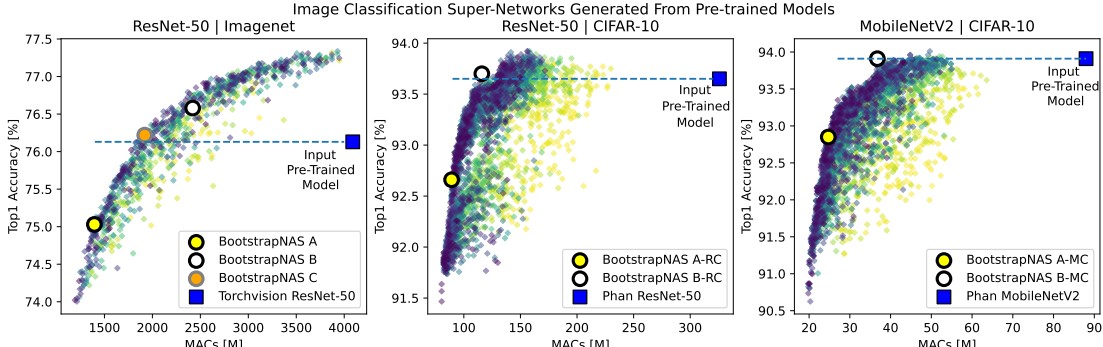

Figure 5: Sub-network search progression for three examples of super-networks generated by Boot-strapNAS from pre-trained models: ResNet-50 trained with Imagenet (left) and CIFAR10 (center), and MobileNetV2 trained with CIFAR10 (right). Section 4 includes results for MobileneNetV2 and MobileNetV3 trained with ImageNet, and EfficientNet trained with CIFAR-100. NSGA-II was used to discover a Pareto front. Each rhombus represents a sub-network with a particular top 1 accuracy and MACs. All sub-networks above the dashed lines outperform the pre-trained models given as input in both objectives, top1 accuracy and MACs. NSGA-II was configured to 1,000 evaluations (20 generations) for Imagenet-trained models and 3,000 (60 generations) for CIFAR10-trained models.

| Input Models Trained with CIFAR-10 from (Phan, 2021) | Pre-Trained Top 1 Acc. [%] | Pre-trained MACs [Millions] | ISO-Top 1 Acc. BootstrapNAS (Fewer MACs) | Drop ~ 1% Top 1 Acc. BootstrapNAS (Fewer MACs) |
|---|---|---|---|---|
| ResNet-50 | 93.65 | 325.80 | **2.81×** | **3.65×** |
| MobileNetV2 | 93.91 | 87.98 | **2.39×** | **3.56×** |

Table 2: Improvements obtained by selected sub-networks discovered by BootstrapNAS. We compare two sub-networks for each base model: One sub-network that maintains the accuracy (ISO-Top 1) while reducing the number of MACs, and another one that allows for a drop of ~ 1% in accuracy but with a greater reduction in MACs.

efficient than the given pre-trained model. Figure 5 on the left illustrates the search progression of NSGA-II on ResNet-50. This figure highlights three sub-networks: BootstrapNAS *A* provides a reduction in model size in terms of MACs by 66.1%, with an accuracy drop of less than 1%, while BootstrapNAS *B* reduces MACs by 40.8% with an improvement in accuracy. The third sub-network obtained from the pre-trained ResNet-50, BootstrapNAS *C*, maintains the top 1 accuracy of the input model while reducing the number of operations in MACs by 53.1%.

Improvements are observed for the MobileNetV2 and MobileNetV3 models from Torchvision, as well. As described in Table 3, in the case of MobilenetNetV2, BootstrapNAS discovered a sub-network that reduced the number of required MACs by 12.5% with a minimal drop in accuracy from 71.88% to 71.42%. In the case of MobilenetV3, BootstrapNAS discovered a sub-network that requires 21% fewer MACs than the given model with a minimal drop in accuracy from 74.04% to 73.52%. This improvement is achieved with 25 epochs of super-network training and only enabling the elastic dimension.

We used the selected sub-networks from the ResNet-50 super-network to analyze their performance using a dual-socket Intel® Gold 6252 CPU @ 2.10GHz (Cascade Lake), each CPU with 24 physical cores. To evaluate the latency of the models, we processed a single sample (of size 1), measured the completion time, and calculated the 90th-percentile latency in milliseconds. Bootstrap-NAS *A* provides the largest improvement in latency over the input model with 2.16× improvement

| Input Pre-trained Models | Pre-Trained Top 1 Acc. [%] | Pre-trained MACs [Millions] | Sub-network BootstrapNAS Top 1 Acc. [%] | Sub-network BootstrapNAS MACs [Millions] |
|---|---|---|---|---|
| EfficientNet-B0 (CIFAR-100) | 87.02 | 385 | 86.89 | 338 |
| MobileNet-V2 (ImageNet) | 71.88 | 301 | 71.42 | 263 |
| MobileNet-V3 (ImageNet) | 74.04 | 216 | 73.52 | 169 |

Table 3: Comparison between the pre-trained models (EfficientNet-B0, MobileNetV2 and MobileNetV3) given as input to BootstrapNAS and a discovered sub-network that outperforms the given model after a few epochs of super-network training on the elastic depth dimension.

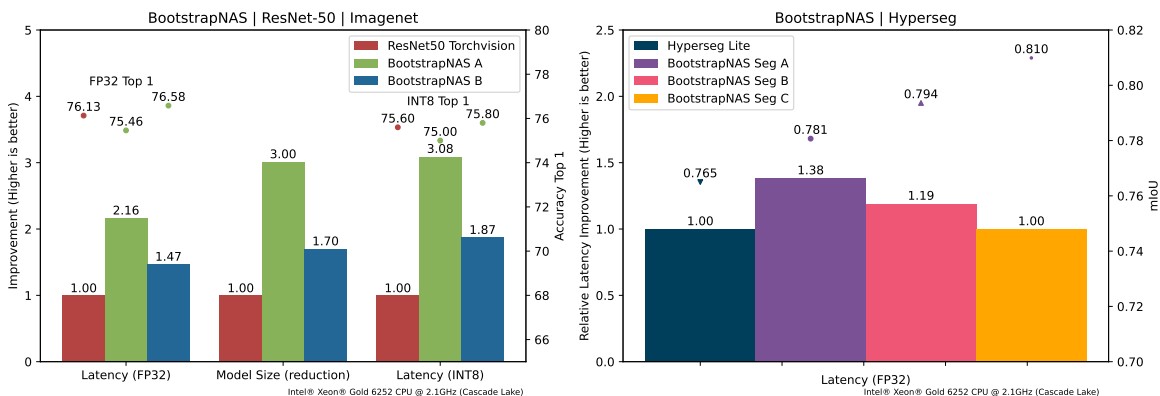

Figure 6: Left: Performance of two examples of sub-networks extracted from the super-network generated from the pre-trained ResNet-50 model from Torchvision. These sub-networks outperform the baseline pre-trained model on latency (both for FP32 and INT8) and model size. Right: Performance of three examples of sub-networks extracted from the super-network generated from the pre-trained HyperSeg Lite model. These sub-networks improve on mIoU, and two of them on latency, over the baseline model.

in FP32 and 3.08× in INT8 as shown in Figure 6 on the left. BootstrapNAS *B* provides the minimal improvement of 1.47× and 1.87× for FP32 and INT8, respectively. To evaluate throughput in samples per second, we used batch processing where the latency is unconstrained and all data is available and processed in any order. BootstrapNAS *A* improves the throughput from the original model by 2.66× in FP32 and 2.57× in INT8. BootstrapNAS *B* and *C* provide comparable performance with improvements in throughput of 1.55-1.98× for both FP32 and INT8. Overall, BootstrapNAS *A-C* in INT8 can achieve a 6.02-9.87× improvement in throughput in comparison to the pre-trained FP32 Torchvision ResNet-50 model.

**Super-Networks from Models trained with the VFX Segmentation Dataset**. We also generated a super-network from a pre-trained HyperSeg Lite model (Rhodes and Goel, 2020). HyperSeg is a model for end-to-end interactive video segmentation tasks for high-resolution (2K) data. HyperSeg outperforms state-of-the-art models for interactive segmentation (Z. Li et al., 2018), DOS (Xu et al., 2016), Graph Cut (Boykov and Jolly, 2001), and Random Walk (Grady, 2006) with a mIoU of 0.840. We used HyperSeg Lite which uses lower resolution data (224x224) and obtains a mIoU of 0.765. Figure 6 on the right shows the improvements obtained by three sub-networks from the BootstrapNAS' super-network: BootstrapNAS *Seg A*, with 1.38× less latency than the baseline, and BootstrapNAS *Seg B*, with 1.19× less latency. Both sub-networks improve on accuracy, 2% and 3.8% respectively in comparison to the baseline model's accuracy. BootstrapNAS also discovered a

sub-network, BootstrapNAS *Seg C*, with similar latency as the baseline but with a 5.9% improvement in accuracy.

## 5 Conclusion

We have presented BootstrapNAS, a framework for neural architecture search based on the automated generation of weight-sharing super-networks from existing pre-trained models. BootstrapNAS supports super-network training applied to various elastic dimensions and the search for high-performing sub-networks. BootstrapNAS has a simple and extensible API to enable custom implementations of super-network training and search algorithms. We demonstrated the feasibility of BootstrapNAS by applying it to public pre-trained models, showing that it can produce sub-networks with substantial improvements in the performance-accuracy trade-off even for lightweight models such as MobileNet and EfficientNet.

**Acknowledgements**. Thank you to Ravi Iyer, Yuri Gorbachev, and Soren Knudsen for their continuous support to this research. Thank you to the NNCF/OpenVINO team for their help integrating this research into NNCF. In particular, thank you to Wonju Lee, Daniil Lyakhov, and Minje Park. We would also like to thank Vui Seng Chua and Yash Akhauri for their invaluable feedback.

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

## A  Additional Details on Enabling Elastic Dimensions

**Elastic kernels** are implemented by intercepting the weight tensors of the selected modules and applying element-wise calculations to these weight tensors. BootstrapNAS performs sub-tensor selection and multiplication with an additional tensor of trainable parameters, as proposed by (Cai, Gan, et al., 2020) to remove the spatial dependency of kernel variants on each other inside a given layer. The resulting tensor of an operation with elastic kernels must have a fixed resolution. This is accomplished by applying, when necessary, a different padding for each kernel size, effectively preserving the output spatial resolution with respect to the input; otherwise, the kernel selection process is not well-defined in terms of the output tensor shape.

**Elastic width**. By analogy, *elastic width* can be enabled by reusing the aforementioned mechanism from NNCF (Figure 2). This wrapping mechanism overrides the layer's implementation in such a way that its parameters used in calculations can be intercepted and updated by an arbitrary operation. In order to reduce the width of the layer, the least important output channels of weights can be cut off. The input channels of the next layers are trimmed accordingly for consistency. This operation can be effectively implemented by conventional tensor slicing, provided that the filters are reorganized in descending order of their importance. BootstrapNAS automatically reorganizes the outputs channels of arbitrary models before starting the super-network optimization stage.

BootstrapNAS assigns elastic width values to elastic layers by taking into account the dependencies between them. For instance, two convolutions are dependent if their outputs are the input of an element-wise operation (e.g., addition, multiplication, or concatenation), so they cannot have a different number of filters at the same time. Otherwise, the element-wise operation cannot be performed on tensors of different dimensions. Therefore, all such layers are combined into independent groups by traversing the execution graph. BootstrapNAS uses these groups/clusters to assign the same width values to all layers in the group.

**Elastic depth**. Figure 2 illustrates the procedure for implementing *elastic kernels* and *elastic width* in the super-network. *Elastic depth* requires a different kind of analysis. The implementation of *elastic depth* implies two actions: (i) detecting the blocks of layers that might be removed from the super-network to generate shallower sub-networks, i.e., elastic blocks (before optimizing the super-network). and (ii) skipping a subset of these detected blocks when sampling sub-networks (in the super-network optimization stage).

The algorithm for detecting blocks that can be skipped relies on the shapes of inputs and outputs for a candidate block. Such blocks must satisfy two conditions: (i) if removed, they should not change the shape of feature maps and (ii) they should have a single input and a single output. If a block has several branches at the input, but identical tensors run along them, then we suppose that the block still has a single input; similarly with the outputs. The block detection algorithm is able to find the building blocks of popular networks, e.g., Bottlenecks for Resnet-50 and Inverted Residual Blocks for MobileNet-v2. However, the list of potential blocks that can be removed might be too large. For example, even consecutive Convolution, BatchNorm and ReLU may produce six blocks: [Conv], [ReLU], [BN], [Conv + BN], [BN + ReLU], [Conv + BN + ReLU]. To avoid this excessive generation of candidate blocks, we combine convolution, batch normalization, and activation layers in the graph into a single node and perform the search for elastic blocks on such a modified graph.

A large number of nested blocks are eliminated by discarding the ones that are a superposition of other blocks.

NNCF is capable of bypassing the elastic blocks that have been selected to skip, effectively allowing for the generation of sub-networks of different depths. A group identifier, $g_i$, is assigned for the sequence of blocks. This identifier is used during the super-network optimization stage to filter blocks that should not be skipped or that result in invalid sub-network configurations.

## B User Interaction

When developing BootstrapNAS, we have paid special attention to the usability of the framework. Non-experts should be able to effortlessly convert their pre-trained models into super-networks that can be optimized using state-of-the-art Neural Architecture Search. As illustrated in the code sample below (Listing 1), BootstrapNAS abstracts away all the complexities of transforming the pre-trained model into a super-network, allowing the user to generate a super-network with a few lines of code. Once the super-network has been generated, a selected search algorithm finds high-performing sub-networks.

```
# Produces a model suitable for elasticity.
nncf_network = create_nncf_network(model, config)
# Creates an algorithm for super-network training and adds elasticity.
training_algorithm = EpochBasedTrainingAlgorithm.from_config(nncf_network, config)
# Trains the super-network.
super_network, elasticity_ctrl = training_algorithm.run(
    train_epoch_fn, train_loader,
    validate_model_fn, val_loader,
    optimizer, checkpoint_save_dir)
# Creates an algorithm for sub-network search.
search_algo = SearchAlgorithm(super_network, elasticity_ctrl, config)
# Runs the search.
elasticity_ctrl, best_subnet_config, performance_metrics = search_algo.run(
    validate_model_fn, val_loader,
    checkpoint_save_dir)
```

Listing 1: Super-network generation and search for Pareto optimal sub-networks.

Users can incorporate BootstrapNAS' functionality into their custom training pipeline. As the pseudocode above exemplifies, the user's training and validation functions are passed as arguments to BootstrapNAS, which manages the optimization of the super-network using this information.

## C ResNet-50 and BootstrapNAS A Sub-Network Comparison

To understand the differences between the baseline Resnet-50 from Torchvision trained on ImageNet, and the discovered sub-network BootstrapNAS A (Section 4, Figures 5 and 6), we can look at the characteristics of each model. We first analyzed the number of layers per operation. The baseline ResNet50 model contains 344 layers, while BootstrapNAS only contains 224. This is a ~35% reduction in the total number of layers. Specifically, ResNet50 has ~1.3x more Scale, Bias, Convert, Subtract, and Convolution operations than BootstrapNAS A. Convolutions require the most compute from these operations and this difference directly contributes to the 2.95x difference in multiply-accumulate (MAC) operations, which impacts the latency. The kernel size used for each

convolution also contributes to the amount of compute. Both models use a 7x7 convolution on the input image, but the differences are in the number of 1x1 and 3x3 kernels. BootstrapNAS can obtain comparable accuracy with 22.2% reduction in 1x1 kernels and 25% reduction in 3x3 kernels.

## C.1 Configuration Files

Listing 2 is an example of a configuration file that can be used with BootstrapNAS. Next, we discuss a few important fields that can be used to tune the behavior of BootstrapNAS. An exhaustive list of all the available properties is in a schema file for BootstrapNAS that is used to validate configuration files.

**model**. Specifies the name of the model. This is used to load one of the models known to NNCF. The user can specify their own models, as well.

**progressivity_of_elasticity**. Specifies the order in which elastic dimensions must be applied when using the progressive shrinking training strategy.

**batchnorm_adaptation**. Specifies the parameters used for batch normalization adaptation, e.g., the number of samples used to reset the batch normalization statistics.

**schedule**. This field specifies how the training strategy must be applied. As exemplified in Listing 2, it contains a list of stage descriptors. For instance, in the first stage, BootstrapNAS focuses only on elastic depth. There are several properties that can be specified for each stage. For instance, the user can indicate the level of elasticity in the depth dimension (depth indicator), i.e., the maximum number of layers/blocks that could be removed from each group of layers. BootstrapNAS uses this information to validate elastic depth configurations requested by the training algorithm. The user can also specify how many epochs must be used for each stage, whether to reorganize weights in order of importance, the maximum number of possible values allowed for elastic width configurations (per layer) in a particular stage (width indicator), whether we should apply batch normalization adaptation, and the learning rate for the stage. BootstrapNAS uses cosine learning rate decay by default. In addition to specifying the initial learning rate for a stage, the user can specify the number of epochs that should be considered in the learning rate decay calculation.

**elasticity**. This field specifies how the selected dimensions should be made elastic. In the example on Listing 2, we have selected width and depth. For each dimension, there are a few properties that can be specified. For instance, the user can specify how to construct the possible width configurations (maximum number of possible values to generate sub-network configurations, the minimum possible width, either to use a step for determining the possible values, or a width multiplier). In the case of elastic depth, the example sets its mode to *auto*, which means that BootstrapNAS will automatically detect all the blocks that could be skipped to generate alternative sub-network configurations. If the user sets this value to *manual*, the user can limit which blocks must be made elastic.

**search**. In the example on Listing 2, NSGA2 has been selected as search algorithm. Additional parameters for the search algorithm can be specified, e.g., number of evaluations, population size, and reference accuracy (other search parameters are listed in BootstrapNAS' schema). As with other fields, if the user does not specify their value, BootstrapNAS will use default values.

```json
{
    "model": "resnet50_cifar10",
    "batch_size": 64,
    // ... Additional fields, e.g., number of classes, dataset, etc.
    "optimizer": { // ... Optimizer-related configuration
    },
    "bootstrapNAS": {
        "training": {
            "algorithm": "progressive_shrinking",
            "progressivity_of_elasticity": ["depth", "width"],
            "batchnorm_adaptation": {
                "num_bn_adaptation_samples": 1500
            },
            "schedule": {
                "list_stage_descriptions": [
                    {"train_dims": ["depth"], "epochs": 25, "depth_indicator": 1, "init_lr": 2.5e-6,
                    ↪   "epochs_lr": 25},
                    {"train_dims": ["depth"], "epochs": 40, "depth_indicator": 2, "init_lr": 2.5e-6,
                    ↪   "epochs_lr": 40},
                    {"train_dims": ["depth", "width"], "epochs": 50, "depth_indicator": 2,
                    ↪   "reorg_weights": true, "width_indicator": 2, "bn_adapt": true, "init_lr":
                    ↪   2.5e-6, "epochs_lr": 50},
                    {"train_dims": ["depth", "width"], "epochs": 50, "depth_indicator": 2,
                    ↪   "reorg_weights": true, "width_indicator": 3, "bn_adapt": true, "init_lr":
                    ↪   2.5e-6, "epochs_lr": 50}
                ]
            },
            "elasticity": {
                "available_elasticity_dims": ["width", "depth"],
                "width": {
                    "max_num_widths": 3,
                    "min_width": 32,
                    "width_step": 32,
                    "width_multipliers": [1, 0.80, 0.60]
                },
                "depth": {"mode": "auto"}
            },
        },
        "search": {
            "algorithm": "NSGA2",
            "num_evals": 3000,
            "population": 50,
            "ref_acc": 93.65
        }
    }
}
```

Listing 2: Example of a configuration file to generate super-networks and search for high-performing sub-networks.

