# OpenReview forum: "Automated Super-Network Generation for Scalable Neural Architecture Search"
_automl.cc/AutoML/2022/Track/Main — AutoML-Conf 2022 (Main Track)_

### Official Review · Reviewer_qsNa · 2022-03-30

**Potential Impact On The Field Of Automl Rating:** 3
**Technical Quality And Correctness:** The paper is technically correct and …
**Technical Quality And Correctness Rating:** 4
**Clarity Rating:** 3

**Summary Of Contributions:**

The authors present a method and framework for automatically constructing a super-network / one-shot model from a pre-trained architecture, overcoming the need for carefully and manually defining one-shot models. This is achieved by varying the given, pre-trained architecture across 3 dimensions: depth, filter and kernel sizes.
The authors build upon the Network Compression Framework (NNCF) (Kozlov et al., 2020).
Once the super-network is build, different strategies can be employed to train the super-network and to search for optimal architectures within the super-network.  The authors additionally consider multi-objective optimization and search for architectures that are optimal w.r.t., e.g., accuracy and latency. For this, off-the-shelf optimizers such as NSGA-II can be used once the super-model is trained.

The authors evaluate their framework with various pre-trained architectures (ResNet, MobileNetV2, V3, EfficientNet) and on multiple problems (image classification for Imagenet and Cifar10 + video segmentation). The proposed framework consistently improves over the baseline architectures, either in terms of accuracy or efficiency or both.

**Clarity:**

The paper is overall well-written and easy to follow.

Some minor comments:
- The experiments section would benefit from additional structure (e.g., subsections) to make it easier to understand what the different experiments are about.
- Figure 6 seems a bit of an overkill; a table might be a better way of representation here; putting all 3 architectures in one figure might also be an option

**Overall Review:**

+ practical framework for automatically constructing super-model given some baseline architecture
+ addressing problem of high practical relevance with little prior work
+ different strategies for training super-model and searching can be employed --> flexible and adaptable framework

- super-network is a bit limited as only depth, filters, kernel sizes are varied (see review summary for more details)
- no major methodological contributions

A study on the computational costs for training and searching the super-model (relative to the cost of pre-training the baseline architecture) would be interesting. What would happen if the baseline architecture is not pre-trained? How would this affect the computational cost and performance of the method?

Minor comment: the reference for the 'once for all' paper (Cai et al.) seems broken. In general, citation style is a bit inconsistent and unclean.

**Potential Impact On The Field Of Automl:**

The paper address an important problem in the NAS literature: how to construct a super model and how to build super models given some baseline architecture (which is a common use-case in practice). The authors propose an interesting and generally applicable approach to tackle this problem.

**Reproducibility:**

The list is filled out in a reasonable way and code is appended, which I briefly looked at. Thus I'm confident that the results can be reproduced.



**Review Confidence:**

4: You are confident in your assessment, but not absolutely certain. It is unlikely, but not impossible, that you did not understand some parts of the submission or that you are unfamiliar with some pieces of related work.

**Review Rating:**

5: Accept, good paper

**Review Summary:**

The authors propose an interesting framework for automatically constructing a super-model for architecture search given some baseline architecture. The framework seems flexible, easy to use and easy to adapt. The authors address an interesting problem and thus the framework could be of great interest for the community.

The only major concern I have is that the super-model is fairly limited since it only varies depth, filter and kernel sizes. Also, the given baseline is the architecture with maximum size within the super-model, i.e., no larger architecture can be discovered by definition. It would be interesting to have more options here, e.g., for also varying the layer types / operations and for also finding larger architectures. E.g., one could employ network morphisms [1,2,3] to expand the baseline architecture. A similar idea as presented in this work is
 employed by [4], which should therefor also be discussed in this work.

Minor final comment: [5] might be worth adding in the discussion on available frameworks in the related work.


[1] Net2Net: Accelerating Learning via Knowledge Transfer, Chen et al.
[2] Network Morphism, Wei et al.
[3] Efficient Multi-objective Neural Architecture Search via Lamarckian Evolution, Elsken et al.
[4] Fast Neural Network Adaptation via Parameter Remapping and Architecture Search, Fang et al.
[5] NASLib: A Modular and Flexible Neural Architecture Search Library, Ruchte et al.

---

### Official Review · Reviewer_8mKb · 2022-04-04

**Potential Impact On The Field Of Automl Rating:** 3
**Technical Quality And Correctness Rating:** 4
**Clarity Rating:** 3

**Summary Of Contributions:**

This work describes BootstrapNAS, a software framework for one-shote NAS. Bootstrap supports the (1) generating and training a supernet based on a pre-trained model (e.g., ResNet-50), (2) and search for the high-performing subnets from the trained supernet. BootstrapNAS also provides APIs for the users to develop their own training strategy, search algorithms, or performance estimators.

The authors provide results on several supernets (e.g., ResNet-50, MobileNetV3, ..) and several tasks (image classification and segmentation) to demonstrate the usefulness of BootstrapNAS.

**Clarity:**

Most parts are clear. But I have confusions about two important details:

(1) This work mentions a lot about using “pre-trained” model to generate supernets. Do we need to re-train the supernets in this case or directly use the network weights from the pre-trained model from the supernet? This needs to be made clear.

I guess the authors still train the supernet since (1) the software supports it, and (2) that should give better weights for supernet. But the term “pre-trained” makes me confused and interpret it as no training of the supernet is involved.

(2) L122 - “hence the transformation … must guarantee that the maximal sub-network results in the same accuracy (within a small margin of error)”. What does this mean?
Does this mean that with the weights directly from the supernet and without any re-training, the maximal subnet needs to have the same accuracy as the given pre-trained model?

If the supernet is trained without using the pre-trained weights, this sounds difficult to me as the supernet weights need to take care of many subnets and might be sub-optimal for the maximal subnet. In practice, we care more about the subnet accuracy after re-training than the subnet accuracy that directly uses the supernet weights.

If the supernet directly uses the weights from the pre-trained model, the accuracy should be exactly the same without marginal error.


**Overall Review:**

Overall I think this is a nice work as the proposed software BootstrapNAS would make it much easier to apply one-shot NAS to existing architectures/search space. The flexibility of the APIs in BootstrapNAS will also make it easier for future research to develop new methods.

My main concerns are about how the “pre-trained” weights are used in BootstrapNAS (see Clarity for more details). This is not just for clarity, as I think they are important technical details that need to be very clear in the paper.

The x-axis in Fig. 6 needs to be re-designed. Now it looks like the  MACs are reduced by more than 10x, although it is just 12% fewer for EfficientNet-B0.


**Potential Impact On The Field Of Automl:**

One-shot NAS is an important NAS method but using it might require extensive engineering efforts for an average deep learning practitioner. This work proposes a software framework to make this process easier. So I think this work can have a good impact on AutoML.


**Reproducibility:**

Yes. Code is provided.


**Review Confidence:**

3: You are fairly confident in your assessment. It is possible that you did not understand some parts of the submission or that you are unfamiliar with some pieces of related work.

**Review Rating:**

5: Accept, good paper

**Review Summary:**

I recommend acceptance due to the usefulness of the proposed software framework.


**Technical Quality And Correctness:**

The technical quality of this work is good. As mentioned in Sec 3, the BootstrapNAS software supports several training strategies of supernet, search algorithms, and performance estimators. They also provide flexible APIs for users to develop their own components. This design is thoughtful and useful.

---

### Official Review · Reviewer_YymL · 2022-04-04

**Potential Impact On The Field Of Automl Rating:** 4
**Technical Quality And Correctness:** I do not find any problems.
**Technical Quality And Correctness Rating:** 4
**Clarity:** This paper is well-written and easy t…
**Clarity Rating:** 4

**Summary Of Contributions:**

This paper introduces a software framework (BootstrapNAS) for supernet-based NAS. Given a pre-trained model, it provides an easy-to-use interface for converting it to a super-net, training the super-net with different algorithms, and searching for good sub-networks under different constraints (MACs, latency). The authors also promise to release the code to the research community.


**Overall Review:**

[pros]
1. This paper is well-written.
2. The authors promise to release their code to the research community.
3. This paper provides extensive experiments on different datasets (ImageNet, CIFAR10, CIFAR100, VFX), showing the generalization ability of their framework.

[cons]
1. This paper does not present new algorithms/methods. But, I think it is fine for a software framework paper.

**Potential Impact On The Field Of Automl:**

I think this paper will interest many audiences of the AutoML Conference. The authors also promise to release their code, which might be useful for many practitioners.

**Reproducibility:**

I think it will not be difficult to reproduce the results, given that the authors will release their code.

**Review Confidence:**

5: You are absolutely certain about your assessment. You are very familiar with the related work and checked all the details carefully.

**Review Rating:**

5: Accept, good paper

**Review Summary:**

Overall, I think this is a good software framework paper that will be useful for many practitioners.

---

### Official Review · Reviewer_NTyQ · 2022-04-30

**Potential Impact On The Field Of Automl:** N/A (Reproducibility Review)
**Potential Impact On The Field Of Automl Rating:** 3
**Technical Quality And Correctness:** N/A (Reproducibility Review)
**Technical Quality And Correctness Rating:** 3
**Clarity:** N/A (Reproducibility Review)
**Clarity Rating:** 3

**Summary Of Contributions:**

N/A (Reproducibility Review)

**Overall Review:**

N/A (Reproducibility Review)

**Reproducibility:**

I found that I was able to reproduce the core work of this paper successfully, though the provided code does not necessarily facilitate easy verification of the paper's major results through reproduction. The provided code and instructions correspond correctly to their description in the Reproducibility Checklist, with the single minor exception of question IV.E. The authors have marked that they did not include new assets in the supplemental material but, in fact, the supplemental material contains a new version of the NNCF library which supports the authors' code.
In the process of attempting to reproduce this work I did encounter a few minor bugs in the provided code. The requirements.txt file cannot be installed as written, as PyTorch 1.9.1 is required as well as torchaudio 0.9.0 which will not work with PyTorch newer than 1.9.0, resulting in a dependency conflict. In the latency predictor notebook, cell 2 must be run before cell 1 as it installs a necessary module for cell 1 and the import for pyplot is missing. In the search and predict notebook "paper_example2.csv" needs to be renamed to "automl_lat_data.csv".
Beyond these bugs, I did not have difficulties running the code provided by following the authors' instructions. These instructions and code, however, were not sufficient to verify that results they produced were comparable to the published results. The search and predict notebook contains code to produce Figures 3 and 4 from the authors' provided data while the latency predictor notebook contains code to train a latency predictor and display its performance in plots fairly similar to the left and center plots of Figure 4. The provided instructions do not make the relationship between these two notebooks clear and the plots in the latency predictor notebook are sufficiently different from those in Figure 4 that I was not able to confirm I had reproduced the results shown in Figure 4 without modifying the plotting code and manually inspecting key values.
The instructions and code provided did not include a process to extract a variety of subnetworks encountered during the search process such as necessary to produce Figure 3 or Figure 5. As a result, I was unable to verify via reproduction that NSGA-II did actually discover the Pareto fronts as depicted in Figure 5. Similarly, though Table 2, Figure 6, and Figure 7 are based on pairwise comparison between the discovered subnet and the input network on the basis of accuracy, MACs, and latency, this information is not made readily available by the provided code or instructions. This makes it difficult to confirm that the provided code actually produces the efficiency improvements described via reproduction.

**Review Confidence:**

5: You are absolutely certain about your assessment. You are very familiar with the related work and checked all the details carefully.

**Review Rating:**

6: Strong accept, should be highlighted

**Review Summary:**

N/A (Reproducibility Review)

---

### Meta-Review · Area_Chair_V3Ki · 2022-05-08

**Recommendation:** Accept
**Confidence:** 5

**Metareview:**

The paper proposes a software framework to help ease the process of training supernets for NAS. The paper also demonstrates the utility of the proposed framework for multi-objective NAS on both classification and dense-prediction tasks. The paper has received 1 strong accept and 3 accept ratings. All the reviewers agree that the paper would be impactful and the software framework would be valuable to the research community. I fully concur with the reviews. The paper is well-written and clear for the most part.

Therefore, I recommend acceptance for the paper.

---

### Decision · Program_Chairs · 2022-05-13

Accept